# Feasibility and Performance of Free-Hand Single-Photon Computed Tomography/Ultrasonography for Preoperative Parathyroid Adenoma Localization: A Pilot Study

**DOI:** 10.3390/diagnostics13132200

**Published:** 2023-06-28

**Authors:** Mélanie Champendal, Mario Jreige, Marie Nicod Lalonde, José A. Pires Jorge, Maurice Matter, Gerasimos P. Sykiotis, John O. Prior

**Affiliations:** 1School of Health Sciences HESAV, HES-SO, University of Applied Sciences Western Switzerland, 1011 Lausanne, Switzerland; melanie.champendal@hesav.ch (M.C.); jose.jorge@hesav.ch (J.A.P.J.); 2Nuclear Medicine and Molecular Imaging Department, Lausanne University Hospital (CHUV), University of Lausanne, 1011 Lausanne, Switzerland; mario.jreige@chuv.ch (M.J.); john.prior@chuv.ch (J.O.P.); 3Visceral Surgery Department, Lausanne University Hospital (CHUV), University of Lausanne, 1011 Lausanne, Switzerland; maurice.matter@chuv.ch; 4Service of Endocrinology, Diabetology and Metabolism, Lausanne University Hospital (CHUV), University of Lausanne, 1011 Lausanne, Switzerland; gerasimos.sykiotis@chuv.ch

**Keywords:** hybrid imaging, SPECT/CT, US, handheld gamma camera, parathyroid adenoma, primary hyperparathyroidism

## Abstract

The aim of this prospective pilot study was to evaluate the feasibility of a new hybrid imaging modality, free-hand single-photon computed tomography/ultrasonography (fhSPECT/US), for preoperative localization of parathyroid adenomas and to compare its performance with conventional ultrasonography and SPECT/CT. Twelve patients diagnosed with primary hyperparathyroidism underwent sequentially US and parathyroid scintigraphy, including SPECT/CT, followed by fhSPECT/US, allowing for real-time fusion between US and freehand-generated gamma-camera images. The fhSPECT/US detection rates were correlated with histopathology, when available, or with the imaging modality showing the most lesions. Based on a per patient analysis, the detection rate was significantly different when comparing SPECT/CT to fhSPECT/US (*p* = 0.03), and not significantly different when comparing SPECT/CT to US (*p* = 0.16) and US to fhSPECT/US (*p* = 0.08). Based on a per-lesion analysis, the detection rate of SPECT/CT was significantly higher than that of US (*p* = 0.01) and fhSEPCT/US (*p* = 0.003), and there was no significant difference in detection rate when comparing US to fhSPECT/US (*p* = 0.08). The main perceived limitations of fhSPECT/US in lesion detection were: (i) lesions localized at a depth ≥4.5 cm; (ii) imperfect image fusion due to tissue compression; (iii) limited spatial manipulation ability of the SPECT mobile camera handheld probe; and (iv) a wide spread of detected activity. In conclusion, clinical use of fhSPECT/US for localization of parathyroid adenomas is feasible, but shows lower sensitivity than conventional modalities and requires technical improvements.

## 1. Introduction

Primary hyperparathyroidism (pHPT) is a common endocrine disorder and the most frequent disorder of the parathyroid glands, affecting women above the age of 50 more often than men [1,2]. In approximately 80–90% of cases, the cause of pHPT is a solitary parathyroid adenoma. Parathyroid adenomas secrete increased amounts of parathyroid hormone (PTH), which causes hypercalcaemia and can lead to osteopenia, osteoporosis, nephrocalcinosis, or urinary tract lithiasis [3,4].

The main treatment for pHPT is surgery, usually a minimally invasive parathyroidectomy in the case of solitary adenomas. Preoperative imaging is indicated to localize the pathological gland to be removed, as well as to identify patients with multiple hyperfunctioning glands who should undergo more extensive surgical exploration of the neck [1]. Several imaging modalities are available for imaging parathyroid lesions, such as single-photon computed tomography coupled with computed tomography (SPECT/CT), F-choline positron emission tomography (PET), ultrasound (US), 4D computed tomography (CT), and magnetic resonance imaging (MRI). In the context of pHPT, US and SPECT/CT are the most commonly used imaging modalities [4,5,6]. US has the advantages of being non-radiating, mobile, readily available, and rather inexpensive, and it provides high-resolution anatomical information. However, US has some limitations: the ability to detect lesions depends on their localization (e.g., lower for ectopic, mediastinal, and deeply located lesions), the experience of the operator, and the structure of the thyroid gland (lower in the case of multinodular goiters) [3,4].

Scintigraphy is the most specific of the available imaging modalities (87%) [7]. Several studies have reported that combined interpretation of US and scintigraphy images (especially with SPECT/CT) improves preoperative localization with “a predictive value close to 100% when the results of the two techniques are concordant” according to Richard et al. [3,4].

Combining two imaging modalities increases diagnostic certainty in both simple cases (i.e., solitary adenoma) and more challenging cases (e.g., intra-thyroidal parathyroid adenomas). This enables the surgeon to plan the operation and make informed decisions regarding whether to perform minimally invasive parathyroid surgery or bilateral neck exploration. Hybrid imaging combining SPECT and CT during the same examination session has been shown to improve diagnostic performance such as sensitivity, specificity, and detection rate in pHPT [8,9].

A novel hybrid imaging modality was recently developed that allows for the fusion of SPECT and US: declipse^®^SPECT Imaging Probe (SurgicEye GmbH, Munich, Germany). Feasibility studies have applied this new hybrid modality to thyroid imaging [10,11,12], lymphatic imaging, and sentinel node investigations [13,14,15,16,17,18,19,20]. Only two case reports describe the use of this method for the detection of parathyroid adenomas [12,21]. To our knowledge, our study is the first to compare this new technology to the standard of care for the detection of parathyroid adenomas, namely SPECT/CT and cervical US.

The purpose of the present study was to prospectively evaluate the feasibility of fhSPECT/US for the preoperative localization of parathyroid lesions in pHPT, and to compare its performance with the currently recommended combination of US and SPECT/CT.

## 2. Materials and Methods

This was a single-arm prospective pilot study to collect quantitative and qualitative data regarding the feasibility and diagnostic performance of fhSPECT/US and compare it to recommended modalities for parathyroid imaging (US and SPECT/CT).

Patients referred for parathyroid scintigraphy from February 2019 to March 2020 were invited to participate. The inclusion criteria were as follows: age older than 18 years, biochemical diagnosis of pHPT (ad minima, PTH, and calcium levels), ability to lie on the examination table for an additional 30 min after the SPECT/CT (three hours in total), and providing written informed consent. Patients could be excluded from study for non-compliance or an inability to provide informed consent. Each enrolled patient sequentially underwent US, parathyroid scintigraphy including SPECT/CT, and fhSPECT/US, allowing for real-time image fusion (Figure 1). Although the fhSPECT and the US are two separate probes, the declipse^®^SPECT Imaging system (SurgicEye GmbH, Munich, Germany) is provided with an infrared tracking system that allows for three-dimensional localisation of the gamma camera and the US probe (by tracking devices placed on the probes) with respect to a reference marker placed on or next to the patient.

To avoid inter-operator variability, the parathyroid US was always performed by the same expert endocrinologist (G.P.S.) using an Affinity 50 instrument (Philips, Amsterdam, the Netherlands) with two linear probes. One probe L18-5 had a better resolution, with a 3.5 cm width and a frequency ranging from 5 to 18 megahertz (MHz), and the other probe L12-5, had better penetration, with a 5 cm width and a frequency ranging from 5 to 12 MHz. In all cases, the US was performed before scintigraphy.

Next, parathyroid scintigraphy was performed after intravenous injection of 550 MBq of Technetium-99m methoxyisobutylisonitrile (Tc-99m-MIBI). This procedure included pinhole imaging approximately 45 min after injection (acquisition time: 10 min), followed by a SPECT/CT performed approximately 1.5 h post injection. The SPECT/CT was performed using a Symbia Intevo^™^ instrument (Siemens Healthineers, Erlangen, Germany) with the following acquisition parameters: 60 30-s images and a 256 × 256 matrix for SPECT, and a dose modulation of 100 kv and slice thicknesses of 2 mm for CT, providing a useful field of view (FOV) of 500 mm.

Finally, fhSPECT/US imaging was performed using the declipse^®^SPECT Imaging Probe (SurgicEye GmbH, Munich, Germany), with a SPECT duration of 1–5 min depending on tracer retention. Following fhSPECT imaging, an acquisition of US images was performed using a GE LOGIQ E9 ML6-15 linear probe (GE Healthcare, Solingen, Germany) with a frequency ranging from 5 to 15 MHz and an acquisition depth fixed at 4.5 cm. These freehand images were captured each time by one of two experienced nuclear medicine physicians (M.J., M.N.L.). The SPECT probe, Crystal Cam CXC-CT40A (Crystal Photonics, Berlin, Germany), was directed in vertical and horizontal planes around the patient’s neck to obtain multiple projections. The specifications of the cadmium zinc telluride (CZT) solid state camera were as follows: low-energy high-sensitivity collimator (LEHS), 9.4 mm spatial resolution. Reconstructions were performed in an iterative maximum likelihood expectation maximization (MLEM) fashion to generate 3D images using a sampling rate of 20 Hz and voxel sizes of 2 mm × 2 mm × 2 mm.

Following the fhSPECT/US acquisition, an additional pinhole image was obtained to analyze Tc-99m-MIBI washout (acquisition time: 10 min). Thereafter, injection of 200 MBq of Tc-99m pertechnetate was performed, and a pinhole image of the thyroid was obtained 10 min after the radiotracer injection (acquisition time: 10 min) [3]. This second injection was part of our usual clinical procedure for a parathyroid scan. Due to lack of a radiopharmaceutical specific to parathyroid tissue, we performed two different radiotracer injections, the first one (Tc99m-MIBI) binding to the mitochondria of both parathyroid and thyroid tissue, and the second one (Tc99m-pertechetate) penetrating only the thyroid tissue. Thus, a visual comparison of the images allowed us to visualize the parathyroid tissue.

The number and location of suspicious parathyroid lesions visualized on the first cervical US were indicated in the endocrinologist’s report, as is routine in clinical practice. The planar and SPECT/CT scintigraphic images, as well as the fhSPECT/US images, were viewed by two experienced nuclear physicians. The number and location of suspicious parathyroid lesions viewed on images were determined by consensus.

Based on the results of the conventional imaging procedures (US and SPECT/CT), the patients were referred for surgical resection of their respective parathyroid lesions.

The data collected for each patient included: age; sex; height; weight; Tc-99m-MIBI injected activity; images and results of US, SPECT/CT and fhSPECT/US, such as localization of parathyroid lesion (positive, negative, suspicious, or multiple lesions), location of lesion(s) and lesion size; histopathology results; pre- and post-surgical laboratory variables such as PTH and calcium levels; and issues related to image acquisition.

The study protocol was approved by the Vaud Cantonal Commission on Ethics in Human Research (CER-VD, protocol number 2018-02085).

All statistical analyses were performed using Stata/IC software (College Station, TX, USA) version 15.0. Sensitivity, specificity, and accuracy were calculated and compared between the different modalities, using the results from histopathology as gold standard when available, or otherwise using the results of the conventional imaging modality (US or SPECT/CT) showing the highest number of lesions in the respective patient. Pairwise comparisons of the different modalities were carried out using the non-parametric McNemar test for dichotomous paired samples. For all statistical tests, a *p*-value < 0.05 was considered significant.

## 3. Results

### 3.1. Patient Characteristics

The characteristics of the cohort are summarized in Table 1 and individual patient data are available in the Appendix A

### 3.2. Detection Capacity

Across all imaging modalities, 16 lesions were detected in the 12 patients. US detected a total of 10 lesions in 10 patients, SPECT/CT detected 16 lesions in 12 patients, and fhSPECT/US detected 7 lesions in 7 patients (Table 2). At the time of publication, 9/12 patients had undergone a parathyroidectomy. According to the gold standard (surgery when available, otherwise SPECT/CT or US results), 17 lesions were detected. Neither US nor fhSPECT/US detected lesions that were not detected by SPECT/CT.

US detected 59% (10/17) of the lesions in 83% (10/12) of the patients. SPECT/CT detected 94% (16/17) of the lesions in all patients (100%). fhSPECT/US detected 41% (7/17) of the lesions in 58% (7/12) of the patients.

On a per patient analysis, the detection rate was significantly different when comparing SPECT/CT to fhSPECT/US (*p* = 0.03), and not significantly different when comparing SPECT/CT to US (*p* = 0.16) and US to fhSPECT/US (*p* = 0.08). On a per-lesion analysis, the detection rate of SPECT/CT was significantly higher than that of US (*p* = 0.01) and fhSEPCT/US (*p* = 0.003), and no significant difference in detection rate was found when comparing US to fhSPECT/US (*p* = 0.08). The lesions detected were primarily in lower positions. Their locations were congruent between different imaging modalities.

In one patient, four hyperactive parathyroids were detected using SPECT/CT, whereas only two were found using US and one using fhSPECT/US. The surgeon decided to remove first the two lesions on the right side, which were detected by both US and SPECT/CT. The left side was not explored because the intra-operative PTH level decreased and normalized upon resection of the two lesions on the right.

### 3.3. Feasibility

The average duration of fhSPECT image acquisition was 188 ± 109 s (range 51–309). Technical issues were encountered when using this novel imaging modality in patients with successful and unsuccessful detection of parathyroid lesions. Indeed, failure of lesion detection by fhSPECT/US in five patients was related mainly to technical limitations. In two of these patients, the lesions were located too deep, which created problems for both US and fhSPECT/US. For example, in patient #1, the lesion, visualized on pinhole (Figure 2a) and SPECT/CT (Figure 2b) images, was located at a depth of 5.2 cm (Figure 2c) This led to a failure of SPECT and US fusion on fhSPECT/US (Figure 2d), as the depth of fhSPECT is limited to 4.5 cm.

In one patient, the radiotracer uptake was visualized within the sternocleidomastoid muscle (Figure 3a) and outside of the patient in an augmented reality representation (Figure 3b).

In the seven patients with detected lesions, some fusions also encountered technical issues; nevertheless, in these cases, detection was still possible despite suboptimal real-time fusion of US and fhSPECT images. One of the issues was the effect of excessively compressing the tissue for US imaging during the fhSPECT/US procedure, which was not the case for the fhSPECT part, as shown in Figure 4.

The second issue observed in patients with detected lesions was the spread of activity detected by fhSPECT (Figure 5). The lesions thus appeared enlarged and less well-circumscribed compared to US or SPECT/CT. Thus, the spatial resolution of fhSPECT was considered inferior to that of the other modalities. The spread was probably due to the suboptimal three-dimension acquisition with random positioning of the hand-held probe and limitations of the handheld sampling angulation (alignment of the detectors, patient’s anatomy, and position).

In one patient, a lesion was not detected on the US, but it was detected using fhSPECT. The location was posterior to the trachea and thus not accessible to US imaging. Nevertheless, due to presence of radioactivity, visualization was possible with the fhSPECT acquisition but was not correlated to an anatomical structure on the fusion image (Figure 6c). The lesion, clearly visible on the pinhole image (Figure 6a), was 12 mm wide and 18 mm long on the SPECT/CT scan and was located at a depth of 3.5 cm (Figure 6b).

Based on the fhSPECT/US real-time fusion image, the lesion measured 7 mm and was located at a depth of 3.2 cm. No US images were available due to a lack of exploration in this anatomical region.

## 4. Discussion

This pilot study evaluated the feasibility of fhSPECT/US in the preoperative localization of parathyroid adenomas in patients with pHPT, as well as its performance compared to stadard-of-care US and SPECT/CT.

### 4.1. Feasibility

The problems observed with the use of this new imaging modality, fhSPECT/US, have also been reported in previous studies [10,15,16,21,22]. These difficulties led to either a lack of detection or to sub-optimal or even aberrant fusion.

The lack of detection could be related primarily to the limitation of the detection depth of the fhSPECT gamma camera due to the limited angle acquisition. This is currently set by device developers at 4.5 cm. This depth is acceptable for “normal” parathyroid gland locations, but it may not be sufficient for some ectopic locations. This was the case in our cohort for three lesions in retro-tracheal or para-esophageal region. This issue has already been noted by Bluemel et al. in sentinel node localization in obese patients, or when the depth exceeded 2 cm [15].

One of the reasons for sub-optimal fusion mentioned in this work and in the literature is the habit of performing tissue compressions for US image acquisition. This compression causes tissue deformation and decreases the location depth of the investigated organ. Two studies proposed the use of pressure transducers to improve this problem and distribute compression evenly [10,22]. The use of elastic registration could resolve this mismatch, but this operation requires a processing time that makes it incompatible with real-time imaging methods such as fhSPECT/US. More simply, it should be acknowledged that use of excess force that causes tissue distortion is unnecessary for cervical US, causing patient discomfort. In this case, excessive force should also be avoided when using fhSPECT/US for parathyroid lesion imaging. Simple contact of the US probe with the gel and the skin under its own weight is sufficient for cervical imaging.

Movements of the patient or of the locator that allows for infrared tracking may also explain suboptimal fusion [10]. De Bree et al. proposed a support to immobilize the patient’s head [16]. Voluntary movements of the patient can thus be limited by good patient coaching, a comfortable position, and the use of a head support. Such measures were not applied in our study, thus misplacement of the patient locator could have occured. The patient locator should preferably be placed on a rigid surface such as the examination table and not on the patient’s cushion or body, as the patient’s respiratory movements can create an artefact upon fusion of the fhSPECT and US images.

Spatial resolution is often described as insufficient in studies on sentinel node localization. Difficulties arise in the discrimination between the injection site and the node, or when several nodes are located close to each other [10,15,22]. For localization of parathyroid adenomas, there is usually no need to discriminate between different focal radiotracer uptakes, with the exception of focal uptake due to thyroid adenomas or thyroid cancer, which was not the case in our cohort. However, a spread of activity was observed. This spread can be explained by a lower spatial resolution of the fhSPECT camera compared to that of the US probe. This spatial resolution is influenced by different parameters, such as detector–patient distance, the solid angle of the detector, number of projections made, intrinsic resolution of the detection system, patient body configuration, selected technical parameters such as collimator, photoelectric peak width, and matrix size, as well as reconstruction methods [23,24]. In this study, the distance was always as small as possible and was easily modified because the gamma camera was directed manually.

In the literature, it is proposed to use a pinhole collimator to improve resolution [15]. However, this would require significant development work to make reconstruction possible using these collimators.

It is important to note that for a correct fusion image, the pixel size of both imaging modalities must be identical. However, in this study, the US pixel size was not verified. This parameter can be very variable and depends on the examination depth and matrix size. It is usually within the millimeter range and therefore smaller than that of fhSPECT [25].

We noted difficulties in the acquisition of the fhSPECT due to a limited number of angular projections as well as a restricted solid angle of the device. In several patients, the limited number of angular projections led to a spread of activity, and in others it led to an aberrant projection on the US image. Both parameters negatively influenced spatial resolution at greater depths, resulting in spreading of the signal and a less precise localization [24]. An increased number of projections seems essential. However, this increase is directly proportional to the duration of the exam. Therefore, a reasonable time must be ensured to avoid motion artefacts.

In addition, manual scanning of the neck using the fhSPECT camera is performed randomly and is therefore very operator-dependent. It would be interesting to propose a robotic arm system holding the fhSPECT probe to establish a more systematic and efficient scan, allowing for a greater number of projections. Several authors have tested operator variability and proposed this possibility of a robotic arm to automate and standardize the procedure [26,27,28]. Matthies et al. showed that the main parameters that influence image quality are the number of projections and their direction [26]. When projections are made in three different planes, the resolution is significantly improved. The increase in projections, combined with a standardized scan, decreases operator dependency and improves spatial resolution [26,27,28].

Another proposed improvement would be to substitute the current reconstruction algorithm, maximum likelihood expectation maximisation (MLEM), with an ordered subset expectation maximisation (OSEM) method. OSEM is an accelerated version of MLEM that classifies iterations into subsets to speed up reconstruction. For an MLEM reconstruction with 16 iterations, it is possible to perform an OSEM reconstruction with four iterations and four subsets (which will be four times faster), or with two iterations and eight subsets (which will be eight times faster) [29].

In summary, optimization of fhSPECT for clinical use in parathyroid lesion localization would require several technical modifications of the probe and system design.

### 4.2. Performance

SPECT/CT was the most successful imaging modality in this study, as it detected all the parathyroid lesions in all the patients. Interestingly, the performance of fhSPECT/US was inferior to that of US, even though both imaging modalities used US. This might be explained in part by a difference in experience of the US operators, given that the fhSPECT/US was performed after US and SPECT/CT (Figure 1) and the fhSPECT/US operator had knowledge of the prior exams’ results. In general, experience has been shown to have a significant influence (*p* < 0.0001) on accuracy and differentiation of thyroid lesions from parathyroid lesions [30]. Another likely reason is that the fhSPECT/US locator had three spheres attached to the probe. This can limit the operator’s ability to manipulate it by restricting the possible angulations due to a loss of tracking signal on some occasions when the spheres are superpositioned.

### 4.3. Limitations and Perspectives

This pilot study has several limitations. The limited sample size and the small number of surgeries performed likely impacted our performance estimates. Moreover, only sensitivity could be calculated because it was not possible to calculate specificity in the absence of true negatives. Another limitation was that the fhSPECT/US images were not performed blindly, as physicians were aware of planar and SPECT/CT images, as is done in clinical routine. Thus, the operators were guided in their explorations.

Notwithstanding these considerations, the feasibility of fhSPECT/US as a new imaging modality for the localization of parathyroid lesions was well assessed, and several relevant observations were made. While it is difficult to ensure that these observations were exhaustive, several of them were made repeatedly during the study and were also reported by others in the literature. Conversely, all the issues identified in the literature were also observed in the present study, considering the relative paucity of data with this technique, with a few published studies in general and only two case reports specific to parathyroid lesions [12,21].

The development and use of this new hybrid imaging modality are not intended to replace SPECT/CT. However, it could be proposed in special cases such as claustrophobic patients or patients refusing additional CT irradiation. If, after necessary improvements, fhSPECT/US could replace US in some cases, such as to detect whether there is associated thyroid pathology, it could reduce the number of appointments for the patient and potentially reduce healthcare costs. Finally, once technical issues have been resolved, fhSPECT/US could also provide added value in countries or areas that do not have access to SPECT/CT imaging.

One main limitation of US, in general, is its operator-dependent nature [3,8,31]. Since fhSPECT/US combines two freehand investigator-directed imaging modalities, it would be interesting to study the influence of this factor on performance.

Performance of fhSPECT and US imaging requires the operator to be in a position close to the patient, which may raise concerns about radiation protection. A higher dose to the whole body and hands is inevitable. Nevertheless, in a study on hand dose during thyroid investigations with Tc-99m injected activities between 60 and 80 MBq, operators received an average dose of 9.25 µSv at the extremities per procedure during an acquisition of approximately 20 min [32]. In parathyroid explorations, injected activities are higher, and the biodistribution of Tc-99m-MIBI is different from that of pertechnetate. As parathyroid investigations require the highest activity injection among the investigations studied by fhSPECT/US, it would be interesting to carry out a study to evaluate the doses received by operators.

## 5. Conclusions

To our knowledge, this pilot study is the first to investigate fhSPECT/US in a prospective cohort of patients undergoing parathyroid investigations. The results indicate that its use for this purpose is feasible but requires technical improvements. The technical limitations identified in this preliminary analysis were primarily due to to detection challenges related to depth, excess tissue compression during US, and restrictions in the spatial manipulation of the US probe and SPECT camera. The sensitivity of fhSPECT/US was lower than that of US or SPECT/CT. Different plausible explanations can be proposed, such as detection challenges related to depth, operator-dependent handling of US and fhSPECT, and the presence of locators on the devices that limited the angles of detection. After technical improvements are made, further studies are warranted to clarify the potential role of this hybrid modality in clinical routine.

## Figures and Tables

**Figure 1 diagnostics-13-02200-f001:**
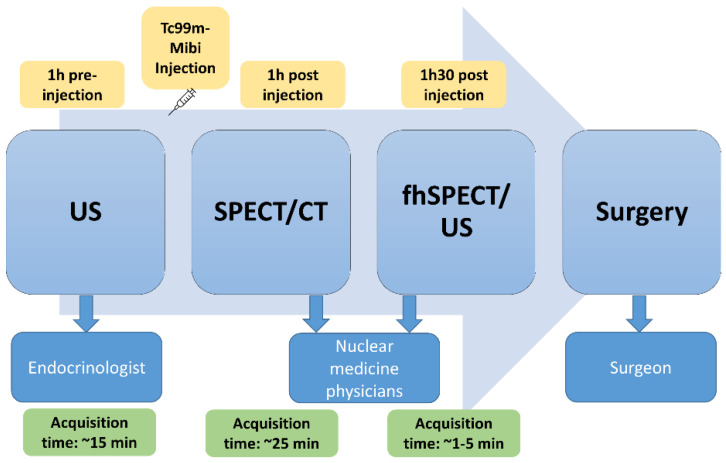
Pathway of patients and medical specialists involved in each step.

**Figure 2 diagnostics-13-02200-f002:**
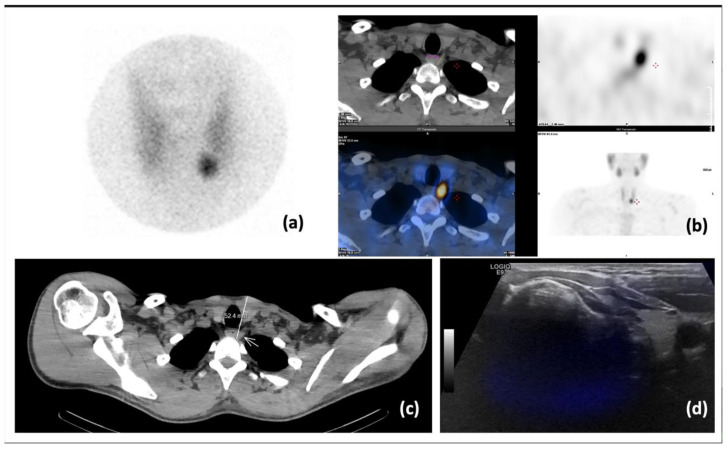
Patient #1: (**a**) pinhole scintigraphy 45 min post-injection of Tc-99m-MIBI; (**b**) SPECT/CT: axial CT (**top left**), SPECT (**top right**) and merged (**bottom left**) slices, supplemented by MIP (**bottom right**). Visualization of left para-esophageal hyperactivity suggestive of a parathyroid adenoma; (**c**) axial slice CT with the parathyroid lesion measured at a depth of 5.2 cm; (**d**) fhSPECT/US was unable to produce a proper fused image.

**Figure 3 diagnostics-13-02200-f003:**
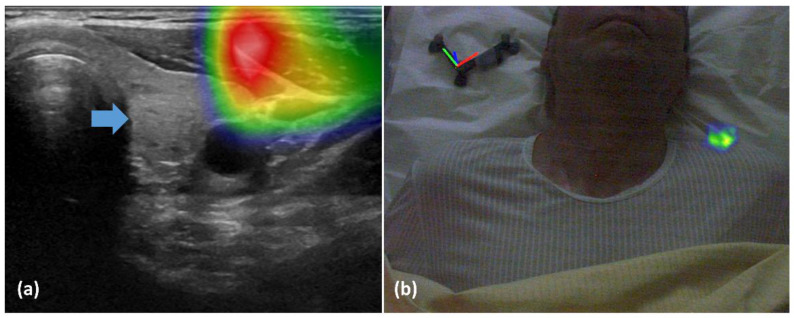
(**a**) fhSPECT/US projects activity within the sternocleidomastoid muscle. The blue arrow indicates the position of the expected radiotracer uptake. (**b**) Augmented reality of the fhSPECT/US shows the activity outside of the patient.

**Figure 4 diagnostics-13-02200-f004:**
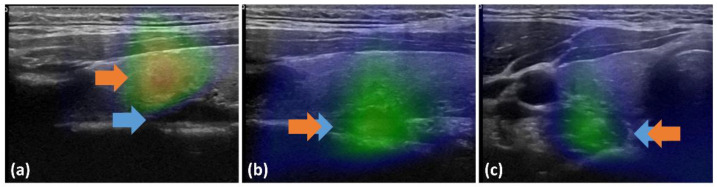
fhSPECT/US fused images. (**a**) When compression is used for the US image acquisition, activity is projected more superficially (orange arrow) to the lesion visible on US (blue arrow). (**b**,**c**) When the US image is acquired without excess force for compression, the activity (orange arrow) is projected on the lesion visible on US (blue arrow).

**Figure 5 diagnostics-13-02200-f005:**
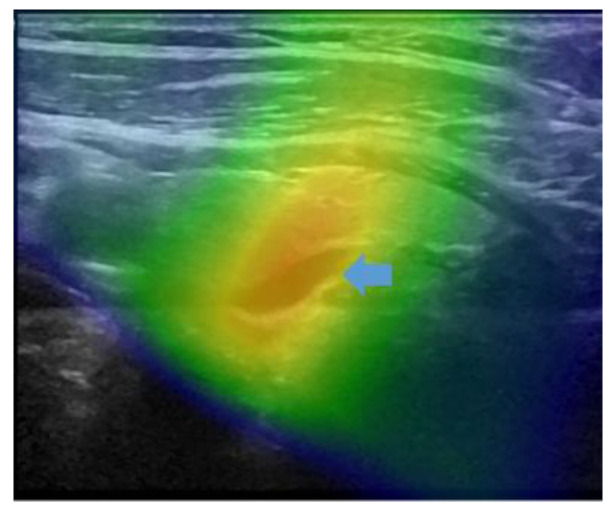
fhSPECT/US mergers showing the spread of activity compared to the size obtained with anatomical images from US. The blue arrow indicates the lesion on the US image.

**Figure 6 diagnostics-13-02200-f006:**
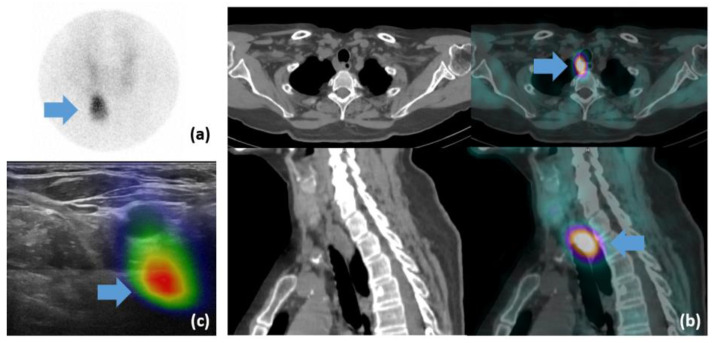
Patient #9: (**a**) pinhole scan 150 min post-injection of Tc-99m-MIBI. The blue arrow shows focal radiotraceur retention. (**b**) SPECT/CT cross-section (**top**) and sagittal (**bottom**) sections of CT (**left**) and SPECT/CT fusion (**right**) images with visualization of retrotracheal hyperactivity (blue arrows). (**c**) fhSPECT/US image: projection and visualization of para-oesophagal metabolic activity with no anatomical structure visible on the US (blue arrow).

**Table 1 diagnostics-13-02200-t001:** Patient Characteristics.

Sex (%)	Age (years) Mean ± SD	BMI (Kg/m^2^) Mean ± SD	PTH Level * Pre-Surgery (ng/L) Mean ± SD [12]	PTH Level * Post-Surgery (ng/L) Mean ± SD [9]	Ca Level ** Pre-Surgery (mmol/L) Mean ± SD [12]	Ca Level ** Post-Surgery (mmol/L) Mean ± SD [10]	Number of Lesions Detected on Imaging	Size of Lesions Measured on SPECT/CT (mm) Mean ± SD	Depth of Lesions Measured on SPECT/CT (mm) Mean ± SD
Female 4 (33%) Male 8 (67%)	60 ± 16	25.5 ± 5.2	107.5 ± 50.9	49.1 ± 29.9	2.7 ± 0.1	2.3 ± 0.9	16	11.4 ± 4.2	33.1 ± 8.9

***** PTH level (normal values: 10–70 ng/L); ** Ca level (normal values: 2.15–2.55 mmol/L).

**Table 2 diagnostics-13-02200-t002:** Number of successful lesion detections by each imaging modality per patient and per lesion.

	Detected (12 Patients)	Undetected According to SPECT/CT (12 Patients)	Undetected According to Surgery (9 Patients)	Undetected According to Gold STD (12 Patients)
**Per patient** (***n* = 12**)				
**US**	10	2	1	2
**SPECT/CT**	12	–	0	0
**fhSPECT/US**	7	5	3	5
**Per lesion** (***n* = 17**)				
**US**	10	6	4	7
**SPECT/CT**	16	–	1	1
**fhSPECT/US**	7	9	7	10

## Data Availability

The data that support the findings of this study are available from the corresponding author, M.N.L. upon reasonable request.

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
