# Peer review of "Feasibility and Performance of Free-Hand Single-Photon Computed Tomography/Ultrasonography for Preoperative Parathyroid Adenoma Localization: A Pilot Study"

_diagnostics, 2023, doi:10.3390/diagnostics13132200_

Round 1

Reviewer 1 Report

The authors demonstrated a pilot study to evaluate the feasibility of free-hand SPECT/ultrasonography for parathyroid adenoma. The authors claim that the proposed scanner shows the feasibility of localization, but the results do not support their claims well. The manuscript should be further improved for publication.

Here are my comments:

1.     The authors mentioned that the free-hand SPECT imaging probe has been investigated in previous studies. Please discuss what is the novelty of this work compared to the previous work. The authors also should summarize the strength and limitations of the previous works.

2.     It seems the ultrasound images were acquired after achieving the fhSPECT images, separately. How the two images were overlaid? How the imaging positions were aligned?

3.     According to Figure 1, the US images were achieved twice (before SPECT/CT and after fhSPECT). What is the difference between the two ultrasound images? It looks the center frequency of the imaging probe is slightly different (11.5 and 8.5 MHz) but I don’t believe this makes a significant difference.

4.     I suggest adding approximate time information, so the readers can understand the protocol. Adding major specifications of each instrument also helps the readers.

5.     Table 1 is too hard to understand. I suggest switching the column and row of the current table. In addition please add the size, depth, and number of lesions, since those parameters affect the diagnostic accuracy, as the author mentioned.

6.     According to Table 2, which is also hard to understand, the results from fhSPECT is the worst in term of diagnostic accuracy. How can we say the probe has feasibility for the localization of lesions?

7.     How the results in Table 2 were acquired? How did you determine (detect) the lesion from each imaging modality? Is there any quantified parameter? If not, how did you calculate p values?

8.     It would be helpful to understand where is the legion in Figures 3-6.

Reviewer 2 Report

Dear Editorial Office

The manuscript I revised is about an interesting topic that is not well known because it is not clinically diffuse.

The proposed procedure has several limitation reported in literature and authors correctely deal with them.

The methods has been weel organized and decripted and authors described precisely all the procedural characteristics also in results.

I agree with authors that affirm this pilot study is a first step in order to know the limitation of the procedure and find solutions.

Considering that i’m not an English native, the text is well written and i didn’t find grammatical or sintax mistakes.

Also figures and tables are well rapresentative and clear.

My suggestion is to accept it.

Best Regards

Round 2

Reviewer 1 Report

The authors have carefully addressed the previous questions in the revised manuscript.  I think the manuscript is acceptable for publication after one minor revision.

In the revised Table 1. Some units are in the first row and some are in the second row. Please be consistent.
